

# GloCAB: Global Cropland Burned Area from Mid-2002 to 2020

Joanne V. Hall[1], Fernanda Argueta[1], Maria Zubkova[1], Yang Chen[2], James T. Randerson[2], Louis Giglio[1]

[1]University of Maryland, Department of Geographical Sciences, College Park, MD, USA
[2]Department of Earth System Science, University of California, Irvine, CA, USA

*Correspondence to*: Joanne Hall (jhall1@umd.edu)

**Abstract.** Burned area estimates are an essential component of inventory-based fire emission calculations, and any inaccuracies in those estimates propagate into the final emission outputs. While satellite-based global burned area and fire emission datasets (e.g. GFED, FireCCI51, and MCD64A1) are frequently cited within the scientific literature and used by a range of users from atmospheric and carbon modelers to policy-makers, they are generally not optimized for cropland

burning – a quintessential small-fire type. Here we describe a new dataset (GloCAB; Global Cropland Area Burned) which represents the first attempt at a global cropland-focused burned area product. The GloCAB dataset provides global, monthly cropland burned area at 0.25° spatial resolution from July 2002 – December 2020. Crop-specific burned area conversion factors for several widespread burnable crops (winter wheat, spring wheat, maize, rice, and sugarcane) were calculated from extensively-mapped cropland reference regions spanning 190,650 fields over 5 different countries. We found global annual

cropland burned area (2003 – 2020) ranged between 64 Mha (2018) and 102 Mha (2008) with an average of 81 Mha using our lower-bound estimates which are substantially higher than the annual average of 32 Mha in the MCD64A1 C6 product. Region-specific trend analysis found some areas with significant increasing trends (northwest India), while the heterogeneity of many other regions found no burned area trends. This cropland-focused burned area methodology is the first step toward improving the representation of global crop-residue burning emissions – an often overlooked small-fire source of trace gas

and aerosol emissions within global fire emission inventories.

## 1 Introduction

Burned area estimates are an essential component of inventory-based fire emission calculations (e.g., Seiler and Crutzen, 1980; UNFCCC, 2022), and any inaccuracies in this component contribute to uncertainties in estimates of emissions and fire impacts on atmospheric composition. Satellite-based global burned area and fire emission datasets are generally not

optimized for cropland burning – a quintessential small fire type – yet these datasets are frequently employed by a broad range of users from atmospheric and carbon modelers to policy-makers in the context of cropland fire monitoring. Reliable remotely sensed fire products are required to quantify trends in fire occurrence and behavior, assess impacts of biomass burning on social and environmental systems, characterize fire behavior and potential future risks, and provide key inputs to fire emission and air quality models. Cropland burned area and associated emissions estimates have been incorporated into

socio-economic decisions, such as economic incentives to reduce crop residue burning (Jack et al., 2022), extension service





programs (e.g., ABC-iCAP, 2022), public health policy (e.g., USDA, 1999), and national emission inventories (e.g., UNFCCC, 2022). Despite the widespread use of cropland burned area and emissions estimates, the methodologies used to produce these estimates are often not well adapted to the complications that are specific to agricultural fires.

Although a single cropland fire has negligible emissions compared to a large wildfire, improved representation of crop-residue burned area and emissions is essential for detection and attribution of trends in air quality in many regions (e.g., Yevich & Logan, 2003; Liu et al., 2021). Post-harvest and/or pre-planting agricultural fires are recurrent, typically occurring once or twice a year (McCarty et al., 2009; Zhang et al., 2018). Due to similar local weather conditions and planting/harvesting cycles in many areas, these fires often occur on or around the same day leading to a pulse of emissions

across a region. Given the proximity of cropland fields to population centers and the magnitude of these emission pulses, health impacts related to exposure from repeated crop-residue burning can be extensive (e.g., Argarwal et al., 2013; Saggu et al., 2018). In addition, despite the low atmospheric injection heights of these burns, crop-residue emissions have the potential to travel great distances and impact locations far beyond the cropland area (Zhou et al., 2018; Hall and Loboda, 2017). For example, Hall and Loboda (2017) found black carbon emissions from crop-residue fires in Russia – as far south as 40°N –

were transported to the Arctic and deposited on snow and ice covered areas. Due to the seasonal timing of the spring pre-planting fires in Russia, the black carbon contributed to the accelerated melting of snow and sea ice (Hall and Loboda, 2017; Hall and Loboda, 2018). Finally, the widespread use of global burned area and emissions products, such as the Global Fire Emissions Database (GFED; van der Werf et al., 2017), within the scientific community (e.g., Friedlingstein et al., 2020; Gao et al., 2018; Kong et al., 2021; Lin et al., 2020) can compound systematic and random errors in cropland burned area

and emissions estimates. The increasing conversion to cropland observed around the world (particularly in Africa; e.g., Brinks et al., 2014; Abera et al., 2020; Li et al., 2019) will likely further increase the need for crop-specific remote sensing approaches that take into account regional and crop-specific differences in management practice and satellite detection efficiencies.

Many factors make accurate satellite-based mapping of crop-residue burned area particularly challenging, including i) the heterogeneity of the global agricultural landscape, ii) the rapid nature of these burns and possible subsequent plowing and/or seeding, and iii) the human-driven changes in fire timing. To effectively map the spatial extent of these small fires, it is ideal to have remote sensing data with high temporal resolution (e.g., less than 2 hours between observations [Hall et al., 2021]), high spatial resolution (less than 10 m to capture pile burns and partial field burns), and appropriate spectral bands. Even

using a combination of Landsat, Sentinel, and Planet imagery would not provide adequate coverage to capture the vast majority of these burns. Furthermore, unlike the 22-year Moderate Resolution Imaging Spectroradiometer (MODIS) record, the longevity and temporal coverage of the Sentinel and Planet platforms is not adequate for long-term studies. Therefore, designing a long-term, global cropland burned area dataset requires a methodology that could balance these requirements while ensuring the magnitude of these fires is adequately captured.




Our new dataset (GloCAB; Global Cropland Area Burned) provides global, monthly cropland burned area at 0.25° spatial resolution from July 2002 to December 2020, encompassing the Aqua and Terra time period. Unlike many remotely-sensed burned area datasets such as MCD64A1 (Giglio et al., 2018) and the FireCCI51 (Pettinari et al., 2021), our methodology does not attempt to directly map burned area. Instead, it aims to estimate the area of cropland burned in each grid cell based

on a combination of MODIS active fire observations and our novel high-resolution database of mapped field-level burned area conversion factors. The active fire product offers both an accurate time of burning and is also able to identify fires that are much smaller than the minimum size that can be mapped with the MODIS burned area product (Giglio et al., 2003; McCarty et al., 2009). Although the Visible Infrared Imaging Radiometer Suite (VIIRS) 375-m active fire product (Schroeder et al., 2014) is even better suited for identifying small fires, the MODIS product has the longevity required for

long-term analyses.

Here we first present a description of the GloCAB methodology, followed by an intercomparison with the MODIS MCD64A1 burned area product (the primary input for the burned area component of GFED), global and regional assessments, and finally product caveats and conclusions.

**2 Data**

**2.1 MODIS Land Cover Product**

The Collection-6 MODIS 500-m Land Cover Type (MCD12Q1; Sulla-Menashe and Friedl, 2018) annual International Geosphere-Biosphere Programme (IGBP) classification product (2002 – 2020) Class 12 (Croplands) and Class 14 (Cropland/Natural Vegetation Mosaic) was used as the base cropland extent. Although other global products exist at a higher

spatial scale (e.g., GFSAD30; Phalke et al., 2020), or are delineated by crop type for a single year (e.g., SPAM; International Food Policy Research Institute, 2019), the annual layers of the MCD12Q1 product have a temporal advantage given the nearly 20-year timespan of this analysis. For this analysis, it was assumed that the MODIS IGBP cropland classes represented the full extent of cropland area each year. As with all remotely-sensed land cover classification data sets, the product contains both omission and commission errors (e.g., Tsendbazar et al., 2016; Zubkova et al., 2023) that represent a

significant source of uncertainty in our regional and global burned area estimates.

**2.2 MODIS and VIIRS Active Fire Products**

The 1-km MODIS Aqua and Terra active fire location product (MCD14ML C6 V3; Giglio et al., 2016) served as the primary input dataset for GloCAB and was obtained from the University of Maryland's SFTP server (sftp://fuoco.geog.umd.edu; Giglio et al., 2020). The MCD14ML product contains multiple variables including latitude, longitude, date, time, and type

for each fire pixel. Active fire pixels were filtered to include only presumed vegetation fires (type = 0) having a center within



500-m of an MCD12Q1 IGBP class-12 or class-14 cropland grid cell. Filtering by pixel-center locations (versus spatially buffered active fire pixel boundaries) helped reduce the double-counting of active fires associated with neighboring land cover classes. The presumed cropland active fire points were summed per month within 0.25° grid cells. To account for the impact of latitude on the sampling frequency, and hence the number of active fires mapped, the monthly active fire counts per 0.25° grid cell ($AF_{count}$) were adjusted using Eq. (1) assuming a reference latitude of 40°N or, equivalently, 40°S (Giglio et al., 2006):

$$AF_{corrected} = AF_{count} \times \frac{\cos(latitude)}{\cos(40°)} \tag{1}$$

The adjusted MODIS Aqua and Terra monthly active fire counts ($AF_{corrected}$) were used in the calculation of the effective burned area per fire pixel per cropland burning reference region (see Sect. 2.4 for details) and the monthly adjusted burned area (Sect. 3.3).

The 0.25-degree, monthly MODIS Aqua and Terra Active Fire Climate Modeling Grid (CMG) Collection-6 products (MOD14CMQ/MYD14CMQ; Giglio et al., 2006) were used to obtain the mean monthly cloud fraction over all cropland land cover pixels. The CMG products contain gridded summaries that include corrected fire pixel counts, mean cloud fraction, and mean fire radiative power.

The Collection-1 375-m VIIRS fire location product from Suomi NPP (VNP14IMGML; Schroeder et al., 2014) was also obtained from the University of Maryland's SFTP server. All active fire locations contain a number of attributes, including the latitude, longitude, date, and UTC time to the nearest minute. The VNPIMG14ML product was only used in the creation of the cropland burned area reference maps (see Sect. 2.4 for details) using the same approach as Hall et al. (2021).

**2.3 Global Crop Type Maps**

We focused on five burnable crop types: winter wheat, maize, rice, spring wheat, and sugarcane. Although there are other crop types that burn (e.g., soybean and cotton), and other agricultural landscapes (e.g., pastures) within each 0.25° grid cell, we focused on these main crop types due to their larger global extent and greater proportion of under-represented fire activity. Global crop type maps were used to assign each 500-m MCD12Q1 IGBP cropland pixel a specific crop type. The crop type data will be used to assign crop-specific burned area conversion factors and emission coefficients (Hall et al., 2022) in later steps, and are an improvement over the generic agricultural waste models often used in previous studies (e.g., GFED4.1s; van der Werf et al., 2017; Randerson et al., 2018).

The GEO Global Agricultural Monitoring (GEOGLAM) Best Available Global Crop-Specific Maps (BACS) are available for several crops, including winter wheat, spring wheat, maize, and rice at 0.05° resolution (Becker-Reshef et al., 2020; Whitcraft et al., 2019). These crop percentage maps are continually updated and are only available as one layer per crop

type. The GEOGLAM initiative is mainly focused on gathering data for the major crop-producing countries participating in
the Agricultural Market Information System (AMIS): United States of America, Canada, Mexico, Brazil, Argentina, EU-28,
Egypt, Nigeria, South Africa, Turkey, Saudi Arabia, Ukraine, Russia, Kazakhstan, China, India, Thailand, Viet Nam,
Philippines, Indonesia, and Australia. These maps use a combination of the best-available remotely sensed crop maps for
each region and crop type. These crop maps were resampled and projected to match the 500-m sinusoidal MCD12Q1 IGBP
cropland layers. The GEOGLAM-BACS does not map sugarcane, therefore the 2010 Spatial Production Allocation Model
(SPAM) global sugarcane physical area (0.08333° spatial resolution) data layer was also resampled and projected to match
the MCD12Q1 cropland layers (International Food Policy Research Institute, 2019).

Any areas where the crop type maps and the MCD12Q1 cropland did not agree were either assigned as a "generic" crop or
were eliminated at the 500-m scale. Specifically, if the MCD12Q1 product labeled a pixel as cropland, and none of the five
crop type maps had an associated crop percentage, then that 500-m grid cell was recorded as a "generic" crop type. These
500-m crop type maps (Figure 1) are used in the calculation of the majority crop type that burns per month per 0.25° grid cell
(Sect. 3.2 for details).

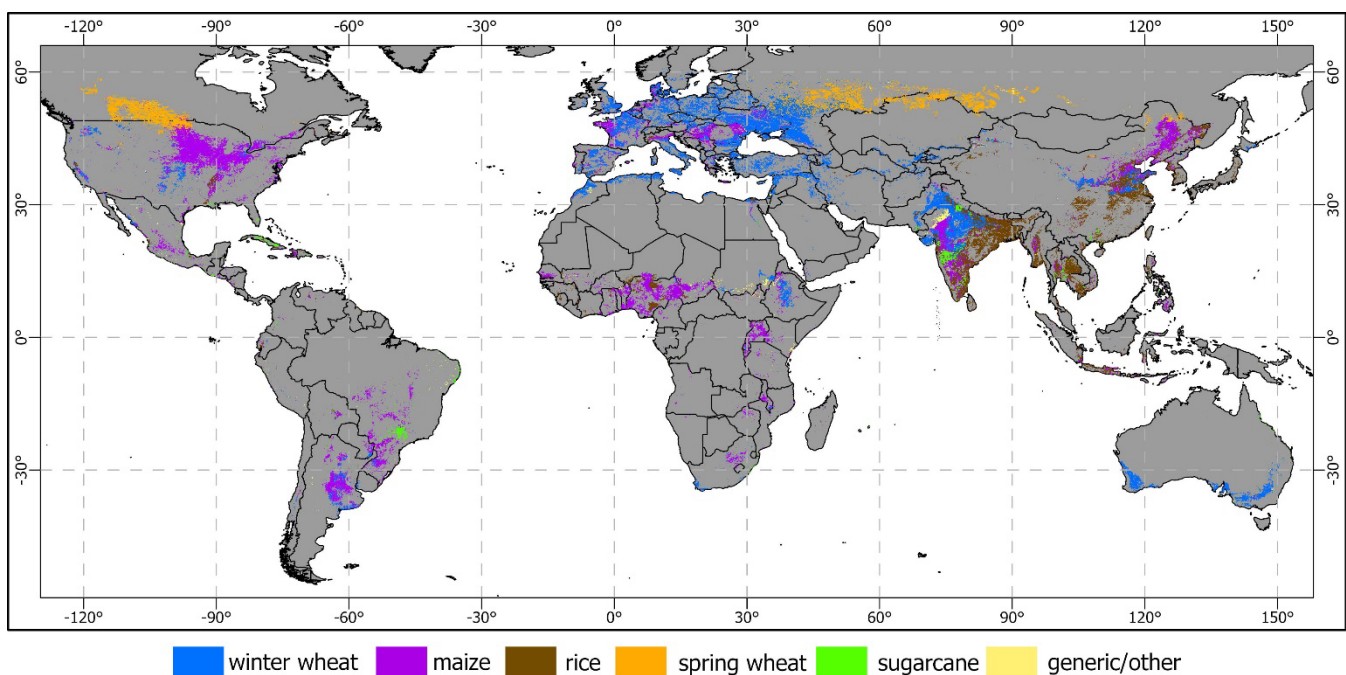

**Figure 1: Example of a 500-m converted crop type MODIS MCD12Q1 cropland layer. The coarser-resolution crop type data from
GEOGLAM-BACS and SPAM were used in the creation of these 500-m annual layers. These annual layers, in conjunction with
the MODIS active fire location product, are used in the generation of the majority crop type that burns per month per 0.25° grid
cell (Sect. 3.2 for details).**



## 2.4 Cropland Burning Reference Areas

Twenty-two cropland burning reference areas were manually digitized and each polygon classified to create highly detailed
cropland field maps over a range of countries, crop types, and dates (Table 1; Figure 2). These reference areas, in
conjunction with the MODIS active fire data, were used to calculate the effective burned area per fire pixel ($\alpha$), which
ultimately serves as a conversion factor that is used to extrapolate our reference areas to much larger regions (Giglio et al.,
2006; see Sect. 3.1). These regions are an extension of previously created cropland field-level burned area maps within
Ukraine (Hall et al., 2021a, Hall et al., 2021b) using a combination of all available 20-m Sentinel-2 Multi-Spectral
Instrument (MSI), 30-m Landsat-8 Operational Land Imager (OLI), and 3-m PlanetScope imagery (www.planet.com), in
conjunction with filtered VIIRS (VNP14IMGML) and MODIS (MCD14ML) active fire location data. The creation of the
reference regions used the more sensitive 375-m VIIRS active fires alongside the 1-km MODIS active fires as an
independent verification for some of the burned fields.

Each digitized polygon was classified via visual interpretation of all available imagery and attributed with the following field
classification: 1 = active flame/smoke or burned area with corresponding active fire polygon (i.e., an overlapping polygon
with a date aligned with the visual change on the field); 2 = definite burned area but with no flame/smoke or active fire
point; 3 = ambiguous (a distinct darkening occurred on the field, but the analysts are unsure if the field was burned then
plowed or only plowed); 4 = definitely unburned; 5 = not cropland or fields are too small that land cover conditions were
difficult to determine on very high resolution (3-m) imagery. Each Class 1, 2, and 3 polygon was also tagged with the
fraction burned (see Hall et al., 2021a for details). For the sake of clarity, the classes will hereafter be referenced using the
following naming convention: definite burn (Class 1 and Class 2), possible burn (Class 3), unburned (Class 4), and non-
cropland/other (Class 5). In total, 190,650 cropland fields were manually digitized by a team of 20 trained analysts and
classified as either a definite burn, possible burn, or unburned.

**Table 1: Summary information on the twenty training reference areas used in the burned area conversion factor analysis and the
two validation reference regions. The locations of reference areas are shown in Figure 2.**

| Reference area (Country_ID) | Mapping start date | Mapping end date | Predominant crop type | Mapped cropland area (km$^2$) | Cropland fields classified |
|---|---|---|---|---|---|
| **Training Regions** | | | | | |
| Brazil_A | 15 Aug 2019 | 15 Oct 2019 | Sugarcane | 1,104 | 4,523 |
| Brazil_B | 1 Jul 2019 | 15 Sep 2019 | Maize | 1,361 | 1,219 |
| Canada_A | 1 May 2018 | 30 Jun 2018 | Spring Wheat | 1,016 | 569 |
| Russia_A | 15 Jul 2019 | 31 Aug 2019 | Winter Wheat | 1,165 | 1,740 |
| Russia_B | 15 Aug 2019 | 30 Sep 2019 | Winter Wheat | 4,601 | 2,295 |
| Russia_C | 1 Apr 2019 | 15 May 2019 | Spring Wheat | 1,309 | 1,116 |
| Russia_D | 15 Apr 2019 | 31 May 2019 | Spring Wheat | 3,362 | 1,362 |
| Ukraine_A | 1 Mar 2017 | 31 Mar 2017 | Maize | 1,498 | 3,995 |
| Ukraine_B | 1 Mar 2017 | 31 Mar 2017 | Maize | 3,569 | 6,168 |
| Ukraine_C | 1 Jul 2017 | 4 Aug 2017 | Winter Wheat | 3,862 | 9,327 |
| Ukraine_D | 1 Aug 2016 | 31 Aug 2016 | Winter Wheat | 2,519 | 5,091 |



| Ukraine_E | 15 Jul 2017 | 15 Aug 2017 | Winter Wheat | 2,480 | 5,433 |
|---|---|---|---|---|---|
| Ukraine_F | 1 Jun 2017 | 27 Jul 2017 | Winter Wheat | 1,300 | 2,758 |
| Ukraine_G | 15 Jun 2017 | 31 Jul 2017 | Winter Wheat | 2,810 | 10,161 |
| Ukraine_H | 1 Jul 2020 | 7 Aug 2020 | Winter Wheat | 38,377 | 123,726 |
| USA_A | 1 Nov 2018 | 31 Dec 2018 | Sugarcane | 74 | 1,091 |
| USA_B | 1 Oct 2019 | 30 Nov 2019 | Sugarcane | 155 | 2,404 |
| USA_C | 15 Apr 2018 | 15 Jun 2018 | Spring Wheat | 1,610 | 1,344 |
| USA_D | 1 Sep 2020 | 31 Oct 2020 | Rice | 294 | 746 |
| USA_E | 1 Sep 2017 | 3 Nov 2017 | Rice | 304 | 1,501 |
| **Validation Regions** | | | | | |
| Russia_E | 1 Oct 2018 | 31 Oct 2018 | Winter Wheat | 404 | 2,614 |
| USA_F | 1 Sep 2017 | 31 Oct 2017 | Rice | 497 | 1,467 |

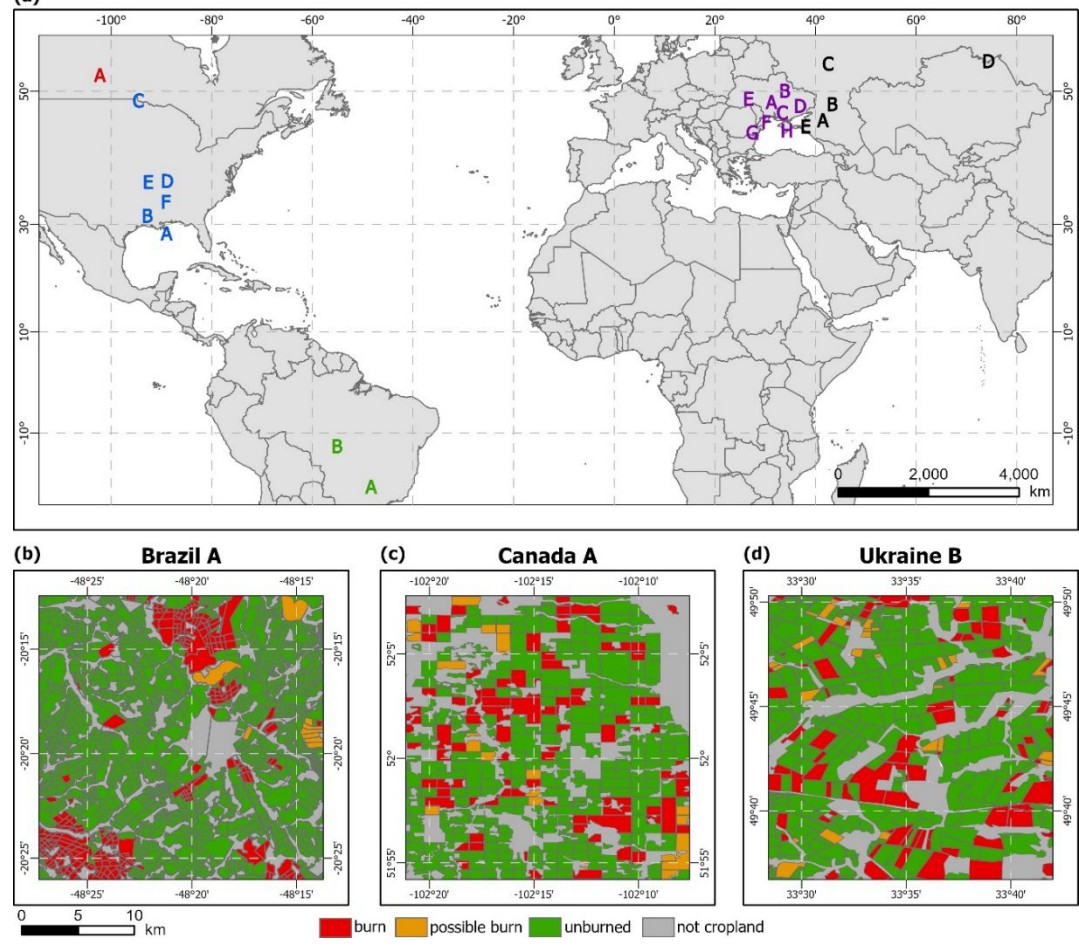

**Figure 2: Location of the twenty-two mapped reference areas (top) with examples of the field-level mapped reference regions from Brazil (bottom left; 15 August 2019 – 15 October 2019), Canada (middle bottom; 1 May 2018 – 30 June 2018), and Ukraine (bottom right; 1 March 2017 – 31 March 2017). These examples illustrate the variety in cropland fields witnessed during the mapping process. For visual distinction, the colored letters (top) represent each individual country: green (Brazil), blue (USA), red (Canada), purple (Ukraine), and black (Russia).**



# 3 Methods

## 3.1 Effective Burned Area Per Fire Pixel

As in Hall et al. (2021a), the conversion factor α was used to extrapolate our high-resolution reference areas to much larger regions. Because our high-resolution reference areas include an indeterminate label (class 3) for fields that could not be unambiguously labeled as burned or unburned, we calculated a lower limit conversion factor ($\alpha_L$) using only the fields with definitive burns (classes 1 and 2) and an upper limit conversion factor ($\alpha_H$) using the definitive burns (classes 1 and 2) and

the ambiguously labeled burned fields (class 3). Each burned field's area for classes 1, 2 and 3 was weighted by its burned area fraction, summed, and then divided by the total number of cropland-filtered MODIS active fire points within the spatial and temporal constraints of each reference area, i.e.,

$$\alpha = \frac{\Sigma(field\ area \times fraction\ of\ field\ burned)}{number\ of\ adjusted\ MODIS\ active\ fire\ points} \qquad (2)$$


Although care was taken to account for different crop burning seasons, crop types, and spatial locations within the twenty training reference areas, several challenges limited the mapping. Most importantly, small fields and poor air quality in several countries (e.g. India and Thailand) prevented the analysts from observing changes in the fields even with 3-m Planet data. Therefore, the final $\alpha_L$ and $\alpha_H$ values for each crop type were based on the median values of a particular combination of

reference regions (see Table 2 and Table 3); the generic crop type was assigned the median value of all twenty training reference regions. We chose the median value over the mean to reduce the impact of any outliers.

**Table 2: Low ($\alpha_L$) and high ($\alpha_H$) conversion factors and the adjusted sum of MODIS Aqua and Terra (A&T) active fire counts for each reference area. The lower limit conversion factor ($\alpha_L$) represents the effective burned area per fire pixel when only including the fields with definitive burns (Class 1 and Class 2), whereas, the higher limit conversion factor ($\alpha_H$) includes both the definite**

**burned fields and the ambiguous fields (Class 3). The locations of the reference areas are shown in Figure 2.**

| Reference area (Country_ID) | $\alpha_L$ (MODIS A&T) | $\alpha_H$ (MODIS A&T) | Adjusted active fire count (MODIS A&T) |
|---|---|---|---|
| Brazil_A | 2.11 | 2.30 | 31.8 |
| Brazil_B | 4.69 | 5.33 | 8.9 |
| Canada_A | 3.92 | 4.55 | 40.9 |
| Russia_A | 4.02 | 4.66 | 46.1 |
| Russia_B | 3.69 | 4.12 | 54.3 |
| Russia_C | 0.29 | 0.40 | 113.9 |
| Russia_D | 7.81 | 8.75 | 45.6 |
| Ukraine_A | 1.84 | 1.96 | 29.5 |
| Ukraine_B | 1.67 | 2.06 | 223.5 |
| Ukraine_C | 1.38 | 1.69 | 226.5 |
| Ukraine_D | 1.56 | 1.80 | 156.2 |
| Ukraine_E | 1.54 | 1.69 | 120.3 |
| Ukraine_F | 2.22 | 2.57 | 91.4 |
| Ukraine_G | 1.84 | 1.96 | 448.2 |
| Ukraine_H | 1.67 | 2.06 | 806.2 |





| | | | |
|---|---|---|---|
| USA_A | 1.02 | 1.31 | 11.3 |
| USA_B | 0.89 | 1.14 | 49.8 |
| USA_C | 2.69 | 7.54 | 20.3 |
| USA_D | 1.85 | 2.25 | 13.8 |
| USA_E | 1.31 | 1.43 | 81.2 |

**Table 3: Low ($\alpha_L$) and high ($\alpha_H$) conversion factors for each crop type used in the calculation of the monthly adjusted burned area.**

| Crop type | $\alpha_L$ (MODIS A&T) | $\alpha_H$ (MODIS A&T) |
|---|---|---|
| Winter Wheat [A] | 1.76 | 2.01 |
| Spring Wheat [B] | 3.30 | 6.05 |
| Maize [C] | 1.45 | 2.70 |
| Sugarcane [D] | 1.02 | 1.31 |
| Rice [E] | 1.58 | 1.84 |
| Generic [F] | 1.76 | 2.16 |

[A] – Median Ukraine (C – H) and Russia (A and B) summer reference areas
[B] – Median Russia (C and D) spring, Canada (A), and United States (C) reference areas
[C] – Median Ukraine (A and B) spring and Brazil (B) reference areas
[D] – Median United States (A and B) and Brazil (A) reference areas
[E] – Median United States (D and E) reference areas
[F] – Median of all reference regions

## 3.2 Monthly Majority Crop Type That Burns in 0.25° Grid Cells

To assign the appropriate value of α to the larger 0.25° grid cell, we first identified the majority crop type (winter wheat, spring wheat, maize, rice, sugarcane, or other/generic) associated with burning – at the 500-m scale – within the grid cell each month. The 500-m sinusoidal crop type maps (Sect. 2.3) were reprojected and resampled to 0.005° to nest inside the larger 0.25° grid cell. The filtered MCD14ML fire pixels were then associated with these 0.005° crop-type grid cells to identify the crop types that burned that month. The majority crop type within a 0.25° grid cell was chosen based on the number of "burned" 0.005° cells. If there was a tie (i.e., more than one crop type had the same number of active fires) then the majority crop type for that 0.25° grid cell was assigned the crop type with the lowest ObjectID (ArcGIS Shapefile database) of those tied classes. This data layer is used to not only assign the crop type α values (Sect. 3.1) but will also be used in a later emissions analysis to assign crop-specific emission factors and combustion completeness.

## 3.3 Monthly Adjusted Burned Area in 0.25° Grid Cells

Monthly burned area was estimated by multiplying the crop-specific α values by the adjusted active fire counts per 0.25° grid cell. Adjustments to these layers were needed to ensure the burned area did not exceed the crop area. Typically, crop-residue burning occurs once or twice a year depending on the crop type and agricultural practices. Depending on the crop type, the two main burning periods occur either before planting and/or after harvest (e.g., Lasko & Vadrevu, 2018; McCarty et al., 2007; Rangel et al., 2018) and often occur in spring and late summer/fall seasons. To account for any double burning within a 12-month period, the final burned area was adjusted to ensure the 6-month cumulative sum (centered on the peak





burning month) did not exceed the crop area within the 0.25° grid cell. The peak burning month was calculated over larger 1° grid cells to identify the month with the largest number of fires on average between July 2002 and December 2020 and this formed the "middle" month of the initial 6-month window. For example, if a 1° cell's peak month was August, the first 6-month window quantified the cumulative burned area between May and October (encompasses the peak in the middle) and

compared that to the cropland area, while the second 6-month window quantified the cumulative burned area between November and April (encompasses a second smaller peak if there was double burning in that region). If there was a tie for peak month, the earliest month was chosen. Quality assessment layers were created to identify the grid cells that were scaled to match the crop area within the 6-month window.

## 3.4 Cloud Cover Analysis

An implicit assumption in estimating burned area with counts of active fire pixels (Sect. 3.3) is that the proportion of fire pixels obscured by cloud (and which were therefore not reported in the MODIS fire product) is approximately the same each burning season. Here it is important to distinguish between raining versus non-raining clouds, and during the cropland burning season we are primarily dealing with the latter, thus for this category of burning clouds must be treated as a source of missed rather than suppressed fires. To understand the potential impact of cloud obscuration on the burned area estimates,

the mean cloud fraction for each cropland grid cell was extracted over the burning season (nominally peak burning month +/-1 month, but in some cases longer). We examined the resulting regional mean cloud fraction (MCF) time series for trends and/or anomalous years that could potentially distort our burned area estimates and trends.

## 4 Results and Discussion

## 4.1 Product Intercomparison and Accuracy Assessment

Validating a cropland burned area dataset requires a more stringent standard than the paired reference-image approach (Boschetti et al., 2009) recommended in the Committee on Earth Observing Satellites Working Group on Calibration and Validation (CEOS) Land Product Validation (LPV; https://lpvs.gsfc.nasa.gov/) protocol. That approach is suitable for more persistent burn scars (e.g., forest fires) but is unsuitable for small and comparatively fleeting cropland fires (Boschetti et al., 2019; Hall et al., 2021b). Given the enormous undertaking required to create the high-resolution reference maps, a large-

scale validation assessment is not feasible. Furthermore, given i) the high omission errors within cropland burned area studies, ii) the inappropriate mapping methodologies typically used for this fire type in previous studies, and iii) the differences in spatial resolution (0.25° in this study compared to 10-30 m resolution in other higher-resolution studies), even a seemingly straightforward product intercomparison is also difficult. Therefore, we conducted several different assessments using two different methods to help understand the accuracy of this product.




For the first accuracy assessment, we compared the manually mapped burned area within two validation regions and the corresponding 0.25° grid cells over the same time period (Figure 3). Given the high cost to create the high-resolution cropland reference regions, our Stage 1 accuracy assessment (Boschetti et al., 2009) was limited to just two 0.25° reference grid cells. Only two out of twenty-two reference regions were chosen for an initial accuracy assessment to ensure sufficient

global coverage for the generation of the GloCAB product. We recognize that this sample is much too small to yield statistically meaningful results, and this proof of concept demonstrates the difficulty in validating cropland burning given the heterogeneity of the landscape. The first validation grid cell was within a predominantly rice area in the United States between 1 September 2017 and 31 October 2017. The average field size within this grid cell was 0.3 km$^2$, which is comparable to the 0.37 km$^2$ average size of fields with the 22 reference regions. The second grid cell was located in a

predominantly winter wheat area in Russia between 1 October 2018 and 31 October 2018 with a much smaller average field size of 0.08 km$^2$. Our validation reference maps found 75 – 84 km$^2$ (using only the definite burn class for the lower estimate) of burned area in the USA grid cell and 69 – 81 km2 burned area in the Russia grid cell, compared to 92 – 107 km$^2$ and 185 – 211 km$^2$ in the respective estimates obtained with $\alpha_L$ and $\alpha_H$. The USA grid cell showed close agreement to the reference burned area, while within the Russian grid cell, the estimated burned area was approximately double the reference burned

area. There are several possible explanations for this discrepancy, including a lack of reference training data within areas with very small fields (i.e., the inability to view the fields using Planet or Sentinel-2 limited our scope in the dataset creation).

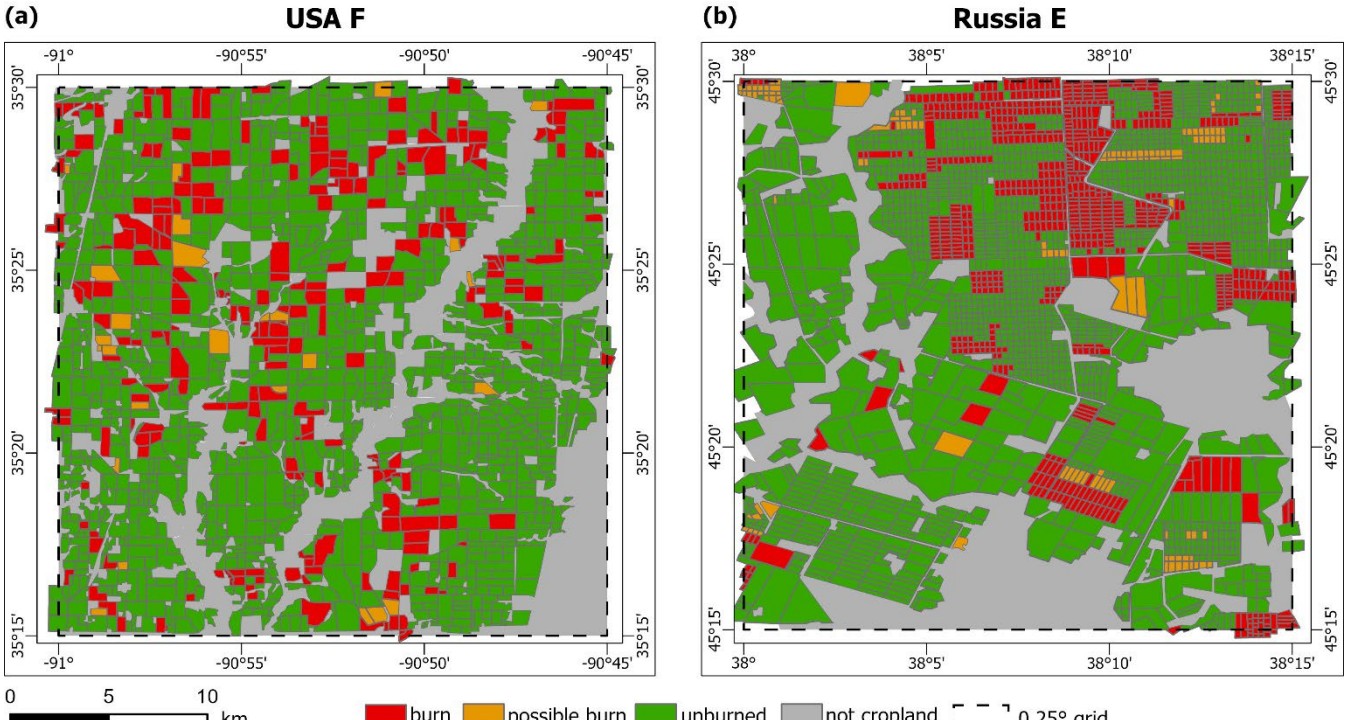

**Figure 3: Maps of the two validation reference regions in the USA (left panel) and Russia (right panel).**




Given that the GloCAB product is designed to be used at a much larger scale than an individual grid cell, we also undertook two regional accuracy assessments in Ukraine and Turkey. The first compared the estimated annual cropland burned area totals for 2016 and 2017 in Ukraine with those calculated in a previous study (Hall et al., 2021b) using higher resolution datasets and a similar methodology but one designed specifically for Ukraine cropland (Table 4). Unsurprisingly, the values in this study's 0.25° output are higher (8 – 11% higher for the lower-limit estimate) compared to the previous study. This is

expected given i) the different reference regions used to derive α values, ii) the coarser 500-m resolution of the MODIS land cover data versus the 10-m land cover map used in the Ukraine-specific study, and iii) the use of MODIS active fires in this study compared to the highly-filtered VIIRS active fire data in the Ukraine-specific study. Visual assessment of both cropland burned area products also show similar spatial and temporal burning patterns – i.e., predominantly maize springtime burning in northern and central Ukraine and predominantly winter wheat burning in southern Ukraine.

**Table 4: Lower and upper limits of 2016 and 2017 estimated annual cropland burned area within Ukraine derived from this study and a previous Ukraine-focused analysis (Hall et al., 2021b).**

| Year | Ukraine cropland burned area ($km^2$) | | | |
|---|---|---|---|---|
| | *Lower Limit* | | *Upper Limit* | |
| | **This Study** | **Hall et al., 2021b** | **This Study** | **Hall et al., 2021b** |
| **2016** | 31,000 | 28,000 | 37,000 | 34,000 |
| **2017** | 42,000 | 35,000 | 50,000 | 45,000 |

The second regional assessment compared the May - November 2019 cropland burned area in the southeastern Anatolia Region of Turkey. Bahsi et al. (2022) estimated cropland burned area through calculating burn severity (difference in

Normalized Burn Ratio, dNBR) using Sentinel-2A/B imagery (10-30m spatial resolution; ~5 day revisit time). Their study estimated 5,100$km^2$ compared to GloCAB's estimates of 5,200$km^2$ (lower limit) and 5,800$km^2$ (upper limit). Given the temporal revisit time of the Sentinel-2 constellation and the propensity for farmers to manipulate their fields shortly after burning (e.g., plowing), it is not surprising that Bahsi et al.'s (2022) values are lower than the GloCAB results. Nevertheless, our cropland burned area estimates are very similar, and given the different methodologies (active fire based versus dNBR-

based), lends credibility to these results.

**4.2 Spatial Distribution and Annual Time Series**

Our study found global annual cropland burned area (January 2003 – December 2020) ranged between 64 Mha (2018) and 102 Mha (2008), with an average of 81 Mha using our lower-bound estimates compared to an annual average of 32 Mha in the MCD64A1 Collection 6 product (Figure 4; Table S1). This 2.7 fold increase in annual average cropland burned area is

unsurprising given the known limitations of the MCD64A1 product within cropland regions (e.g., Giglio et al., 2018; Zhang et al., 2022) and the high omission errors associated with small fires. Breaking these results down by crop type highlights the important contribution of winter wheat and maize burning to global annual cropland burned area, and spring wheat as a contributor to the declining trend.

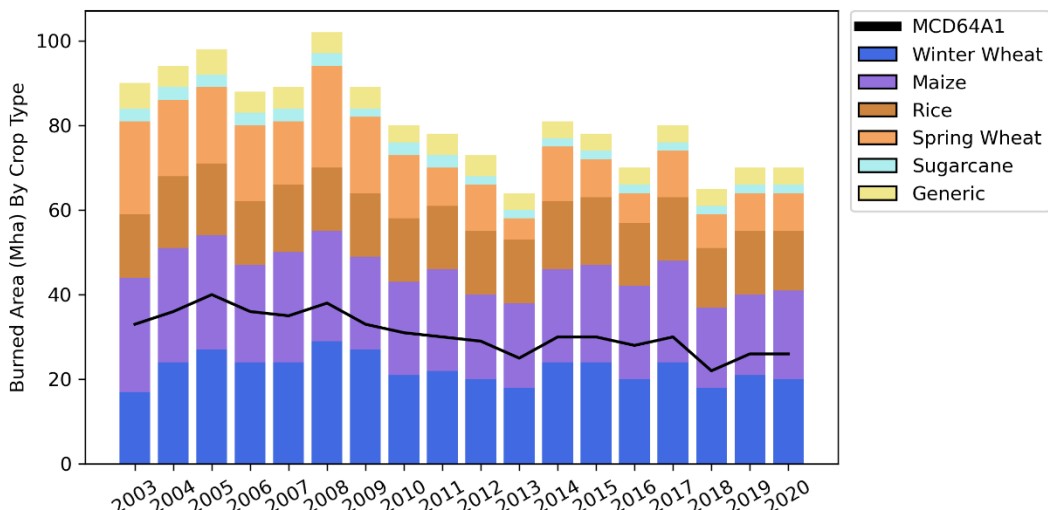

**Figure 4: Annual burned area (Mha) by global cropland burning fire year (June - May) segmented by crop type and overlaid by MCD64A1 (black line).**

Visualizing the annual average burned area (2003 – 2020) both as a fraction of the cropland area within each 0.25° grid cell (Figure 5) and as an absolute area (Figure 6) highlights i) the geographic hot-spots of cropland burning, ii) areas with double-cropping, and iii) areas where neighboring fields with varying harvest cycles are within the same 0.25° grid cell (i.e., fires are recorded in the two crop residue burning seasons but were on different fields). For example, the prevalent wheat and rice crop-residue burning in the double-cropped areas within northern India (e.g., Singh et al., 2020; Sahu et al., 2021), sugarcane burning in Florida and Louisiana, USA (e.g., Hiscox et al., 2015; Sevimoğlu et al., 2019), sugarcane burning in Thailand (e.g., Kumar et al., 2020), and spring wheat burning in central Russia (e.g., McCarty et al., 2012) are all visible. The figures also highlight the prevalence and spatial extent of cropland burning within Ukraine where over 70% of the land area is associated with sown/cropped fields (Hall et al., 2021a). Since the MCD12Q1 IGBP class 12 and 14 pixels also include other agricultural landscapes, certain regions (e.g., Africa) will include more fire activity associated with land clearing and wildfires within pastures as opposed to crop-residue burning. A future refinement will include a separate methodology for non-cropped agricultural landscapes.

Earth System
Open Access Science
Data Discussions

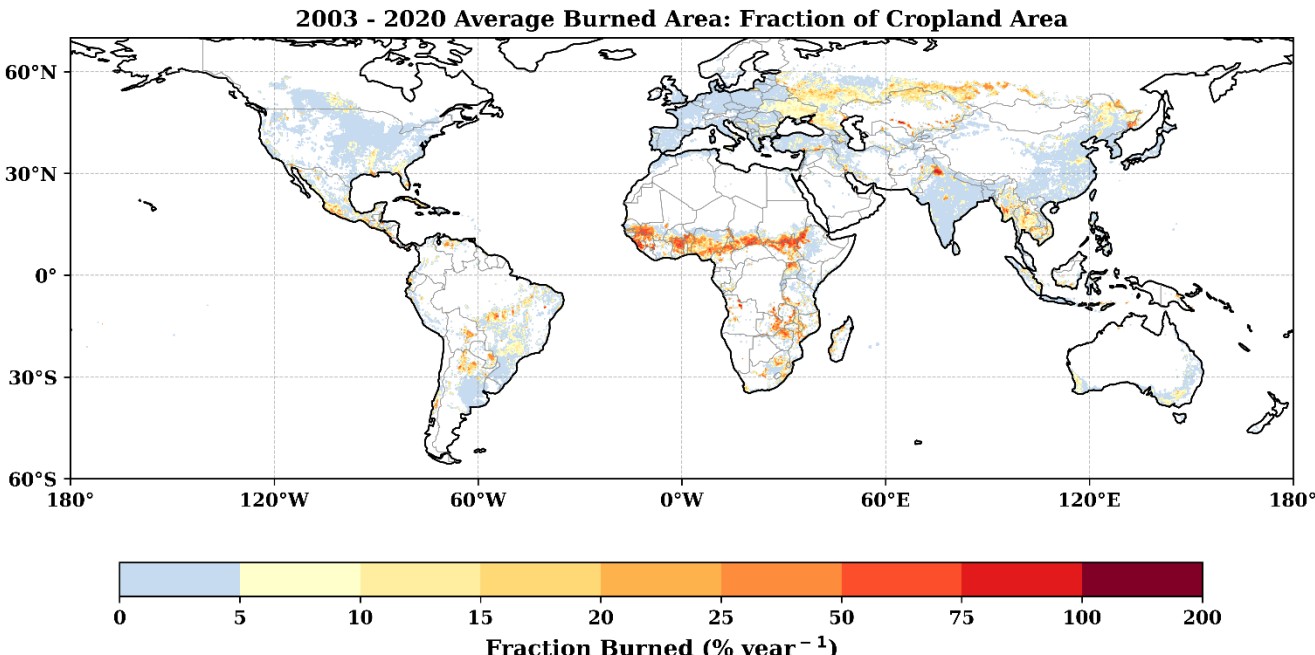

**Figure 5: Average annual area burned (2003 – 2020), expressed as the fraction of cropland within each 0.25° grid cell that burns each calendar year. Grid cells with more than 100% cropland burned area are seen within double-cropping regions or within grid cells where neighboring fields are on different harvest cycles.**

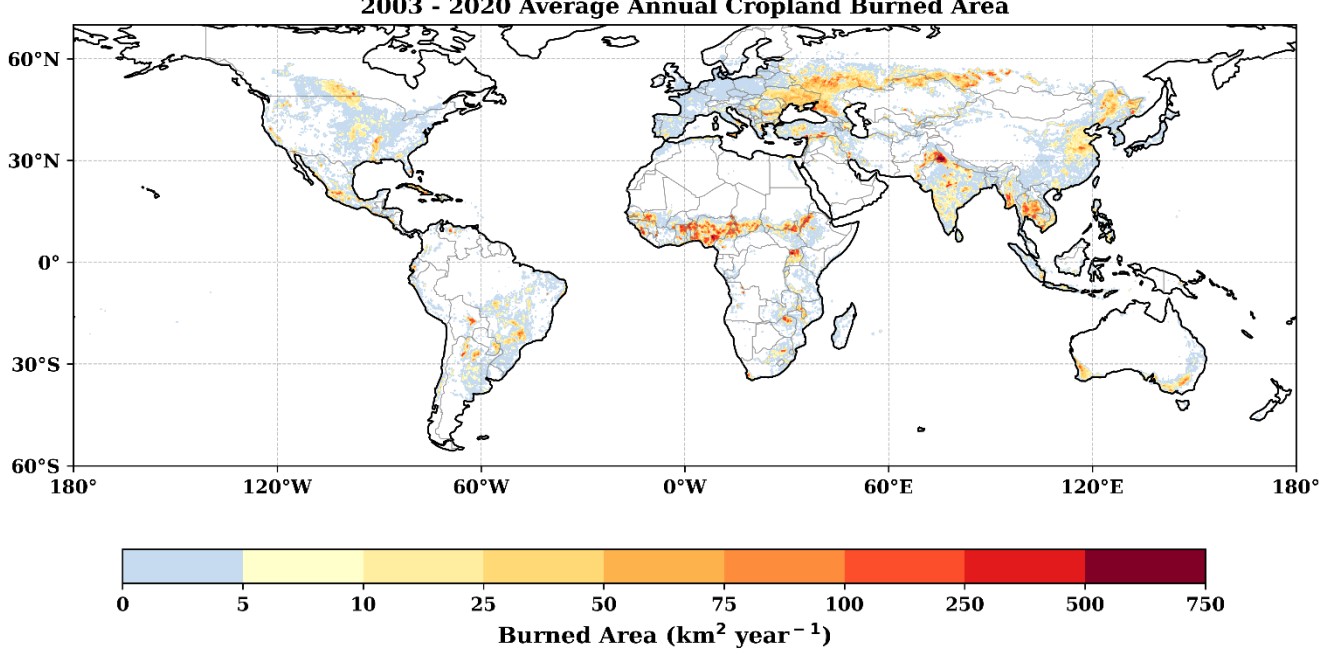

**Figure 6: 2003 – 2020 average annual cropland area burned (units: km²/y) per 0.25° grid cell.**

## 4.3 Cropland Burned Area Trends, Interannual Variability, and Cloud Cover

Analyzing the annual cropland burned area trends requires an understanding of the interannual variability in the timing of the post-harvest/pre-planting burning cycles, the magnitudes of the burned area within broad agricultural regions (see supplementary Figure S1 for monthly burned area trends in the GFED regions), as well as the variability and trends in cloud cover over the peak burning months. Given the complexities (e.g., different crop types, burning practices, and local climates) and heterogeneity within the cropland land cover class, we selected a subset of 6 agricultural regions, including a global extent, representing a range of majority crop types to analyze. For this representative subset, we apply knowledge from a variety of sources including our previous studies (e.g., Hall et al., 2016; Hall et al., 2021a), information gleaned from the 22 manually mapped regions, and scientific and government literature. For each broad agricultural region (and the global estimate) we defined the crop fire year with the month of minimum fire activity indicating the start of the fire year (Giglio et al., 2013). For example, if the month with the least fire activity on average between 2003 and 2020 was September, then the 2003 fire year will run from September 2003 through August 2004. The definition of the cropland fire year will change based on the scale of analysis, and users are therefore encouraged to employ crop fire seasons appropriate for their area of interest. Furthermore, in regions with two distinct cropping cycles, additional trend information can be gleaned by dividing the analysis into 6-month segments – ensuring the two peaks are in the middle of the 6-month window. Finally, only fire years containing all 12 months were used in trend calculations, therefore, some regions contain an extra year of data depending on their monthly burned area distributions.

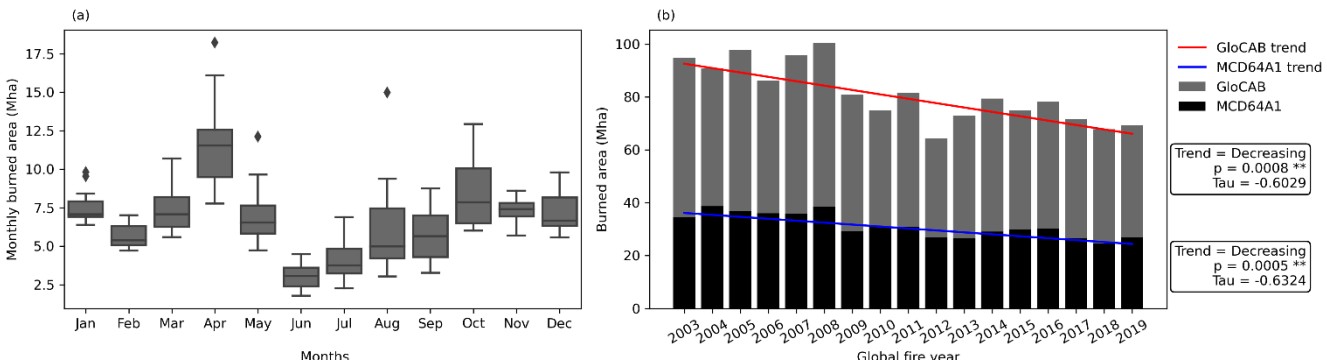

**Figure 7: (a) Global monthly distribution of cropland burned area between July 2002 and December 2020 (units: Mha). Outliers (diamonds) are defined as monthly burned area values greater than 1.5 times the interquartile range above the upper quartile (Q3 + 1.5 × IQR). The monthly median values are represented by the solid line. (b) Global cropland burned area by global fire year (June – May) overlaid with Theil-Sen estimator trend lines with 95% confidence for this study (grey; red) and MCD64A1 (black; blue). Tau represents the ordinal association between two measured quantities.**

Figure 7 shows the global monthly cropland burned area (2003 – 2020) and the annual fire year (June – May) burned area for the lower-limit ($\alpha_L$-based) estimates, alongside the corresponding burned area reported in the MCD64 burned area product, with associated Theil-Sen estimator trend lines and significance statistics (95% confidence). Analysis of the autocorrelation function (ACF) plot and the Durbin-Watson statistic (Durbin and Watson, 1950) confirmed the data were not autocorrelated.



Despite the difference in the magnitude of the two time series, both products show a statistically significant decreasing trend for fire years 2003 through 2019. However, the year-to-year variability in the MCD12Q1 land cover product cropland classes, especially in Africa (e.g., Wei et al., 2020), will have an impact on the burned area trend that is unrelated to the true

burned area patterns (e.g., Verburg et al., 2011; Zubkova et al., 2023). Furthermore, given the limited availability of global crop type datasets, this analysis assumes crop types are constant. Therefore, caution is required when analyzing crop-type land-cover-specific burned area trends by also studying the underlying ancillary datasets within their area of interest to ensure any artificial signals are minimized. Figure S2 highlights the annual global burned area and trends excluding Africa.

Although the globally-aggregated trend is consistent with several other broader studies (e.g., Andela et al., 2017; Arora & Melton, 2018) the heterogeneity within cropland regions is lost at the global scale. Therefore, a regional analysis was undertaken highlighting the differing seasonal cycles and overall annual trends within regional cropland areas (Figure 8). For example, northwest India shows a strong increasing trend and a distinct drop in burned area in the 2019 fire year (August 2019 – July 2020), which likely corresponded to a combination of the Covid-19 pandemic and the 2019 financial incentives

(2,400 rupees; $32 per acre) imposed by India's Supreme Court to help reduce the stubble burning in the northern states (BBC, 2020; News, 2021). In contrast, other regions showed no significant trend. Partitioning the burned area estimates into smaller cropland regions further highlights the difficulty of applying trend lines to these data. For example, the oscillating time series in Ukraine and the abrupt decline in burning in European Russia since 2010 clearly fall outside the domain of a simple linear trend analysis (Figure 8).


Assessing changes in annual cloud cover fraction provided evidence that none of the regional burned area trends were driven by trends in cloud cover or anomalous cloud cover at the beginning or end of the time series. While not a large effect, variations in cloud cover did contribute to the interannual variability in burned area, e.g., in European Russia (with an annual cloud fraction of 30% to 60% during the peak cropland burning months), in 2014 (Figure 9a, bottom) and 2019 (Figure 9b,

bottom).



**Figure 8: (a) Regional cropland monthly burned area (July 2002 – December 2020; Mha) and (b) cropland burned area by region-specific fire year, with respective Theil-Sen estimator trend lines with 95% confidence. Note the difference in vertical axis scales.**
**European Russia had a distinct change in burned area, therefore, the trend lines were subdivided into two periods: 2003 – 2010 (P1; red dashed line) and 2011 – 2020 (P2; blue dashed line).**





It is beneficial to subdivide the trend analysis into crop harvest seasons when further information on the burning of
predominant crop types (for that time period and geographic location) is warranted. Particular care is required when inferring
local patterns with coarse-resolution global products such as GFED and MCD64A1 aggregated to a 0.25° climate modeling
grid. Figure 9 shows the differing magnitudes, spatial patterns, and trends within the contiguous cropland regions of the
United States and European Russia for two periods approximately representing the predominant spring and fall burning
seasons: January – June and July – December. It is clear that the "decreasing trend" in European Russia is driven by the
distinct change in the summer (predominantly winter wheat) burning season compared to the spring burning season
(predominantly spring wheat). This distinct change in burning in 2010 is likely caused by i) the agricultural machinery
deficit between 1990 and 2010, which in turn drove a widespread need to remove crop residue from the fields after harvest
prior to 2010 (Sidorenko et al., 2017), ii) the rise in agro-holdings (i.e., corporate farms) since the early 2000s that led to
large parcels of cropped land no longer being burned and instead efficiently managed with new machinery (Rada et al.,
2017), and iii) the widespread administrative and legal action banning open burning (e.g., Decree of the Government of the
Russian Federation of November 20, 2015, no. 1213; http://government.ru/docs/20511/) after the devastating 2010 fires
(Bondur et al., 2020; Loboda et al., 2017). Figure 8 also shows the seasonal interannual variability within the contiguous
U.S. summer/fall cropland burned area (predominantly post-harvest burning) compared to the relatively stable burning
pattern within the springtime burning seasons (predominantly pre-planting burning).


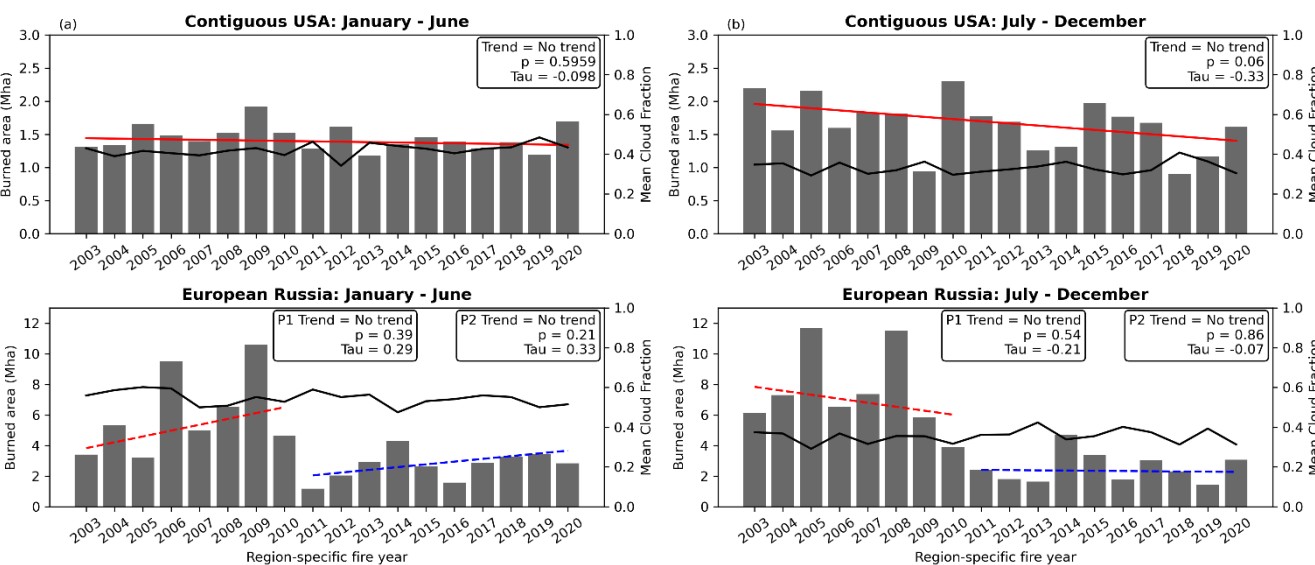

**Figure 9: (a) January – June and (b) July – December cropland burned area time series (grey bars), burned area trends (blue and red lines), and burning-season mean cloud fraction (black lines) for the Contiguous United States (top) and European Russia (bottom). The fire year for the contiguous USA and European Russia runs from January to December. European Russia had a**
**distinct change in burned area, therefore, the trend lines were subdivided into two periods: 2003 – 2010 (P1; red dashed line) and 2011 – 2020 (P2; blue dashed line).**



## 5 Caveats and Limitations

Several caveats and limitations apply to the input datasets and our methodology. First, this study uses static crop-type maps. Higher-resolution, annual crop-type global maps once they become available (e.g., the 30-m Cropland Data Layer for the United States) will improve the identification of cropland areas, including those with either double cropping or crop-rotation practices. For this initial version, the GEOGLAM crop-type maps (https://cropmonitor.org/) were chosen as they are widely used by multiple international humanitarian, government, academic, and research partners and created using the best available data with involvement from local partners within each country (Becker-Reshef et al., 2020; Whitcraft et al., 2019). More accurate and dynamic land-cover data and agricultural-specific maps will help differentiate between field (cropland) and non-field agricultural burning and applying suitable conversion factors and methodologies to each. Further, including a more adaptable partitioning of the crop calendar (i.e., the peak month analysis windows) alongside a more dynamic approach that can adapt to changes in the cropping calendars over time will further refine these results.

Secondly, our results are influenced by the choice of land cover product used as a base cropland extent. While MCD12Q1 was selected since it offers yearly global land cover maps, its coarse resolution might limit the overall accuracy of the estimated cropland burned area, especially within regions where agricultural fields are smaller than MODIS pixel size (500m). Inconsistencies between cropland extents based on various global land cover products in Africa were reported by Zubkova et al. (2023), demonstrating substantial variations in spatial distribution and year-to-year variability. An analysis of springtime burned area in Russia (Glushkov et al., 2021) also highlighted the variability in burned area by land cover class when comparing three separate global land cover products of varying spatial resolutions. Therefore, the availability of higher-resolution yearly land cover products in the future can enhance the proposed methodology, however, the accuracy of those input products will always be an underlying limitation.

Third, initial analysis found the 22 reference regions did not provide sufficient sampling and required too many assumptions for application to the Terra-only period. Additional cropland burned area reference data are required (created using a consistent methodology that is appropriate for cropland burning) over a broader selection of burnable crop types and geographic domains, particularly within areas with predominantly small (average size within 22 reference regions = 0.37 km$^2$) field sizes. Further, the cropland burned area reference data should also span additional years as farming practices change over time and farmers may alter their daily burn times, which could in turn alter the active fire signal. Unfortunately, high-resolution, daily PlanetScope imagery is only widely available from ~2016, as are Sentinel-2 data, thus alternate solutions are needed for the earlier years of the MODIS record. Finally, future work will extend the GloCAB dataset to the Terra-only period from November 2000 – June 2002.



## 6 Data Availability

GloCAB is comprised of lower (low) and upper (high) cropland burned area estimates per month between July 2002 and
440 December 2020 (Hall et al., 2023). The monthly, 0.25 degree GloCAB burned area (units: km$^2$) data are available as annual
GeoTIFF stacks: 12 monthly layers per stack between 2003 and 2020 and 6 layers (July – December) for 2002. The
preliminary GloCAB dataset is publically available on the open repository Zenodo
(https://doi.org/10.5281/zenodo.7860452).

## 7 Conclusions

445 Our new GloCAB dataset provides a global cropland-focused burned area product (0.25°; monthly-time step). GloCAB's
specifications (0.25°; monthly-time step) were selected to match the forthcoming GFEDv5 product (Chen et al., in review),
as GloCAB will be the source of cropland burned area in the updated GFED product. However, GloCAB's specifications are
designed to be useful for other products and applications, and modifications to the methodology can be applied for specific
case-studies. Using twenty exhaustively-mapped field-level reference regions within five countries, the effective burned area
450 per MODIS active fire pixel (α) was calculated for several crop types that are generally associated with burning: winter
wheat, spring wheat, maize, rice, sugarcane, and other/generic. Using these active-fire-to-burned-area conversion factors
(Table 3), we generated lower ($\alpha_L$) and upper ($\alpha_H$) cropland burned area estimates per month between July 2002 and
December 2020. With these data we found the lower-limit global annual cropland burned area (2003 – 2020) ranged between
64 Mha (2018) and 102 Mha (2008) with an average of 81Mha compared to an annual average of 32 Mha in the MCD64A1
Collection 6 product. This increase in annual mean burned area compared to MCD64A1 is unsurprising given the GloCAB
product is specifically designed for cropland burning. Subdividing by crop type also highlighted the substantial contribution
of winter wheat and maize to global cropland burned area. Our analysis also highlighted the heterogeneity within cropland
regions and how burned area trends can be impacted depending on the area of interest, while also discussing the impacts of
cloud cover on burned area totals.


Finally, our next steps include gathering crop-specific emission factors and combustion efficiencies that will be applied to
the monthly majority crop type data layers created in this study. This next step will illuminate the contribution of crop types
to emissions as the proportion of emissions compared to burned area will vary by crop type. Despite the current limitations,
this study set out to develop a crop-specific global burned area methodology that was grounded in high-quality reference
data gathered for this unique fire type. Although burned area reference datasets have recently become available (e.g.,
Franquesa et al., 2020), they are either developed for non-cropland fires, or are sporadic/opportunistic-based on field-work,
surveys, etc. (e.g., Liu et al., 2020; Hall et al., 2016) and are therefore not appropriate for this methodology. In general,
cropland emissions are severely underestimated, primarily due to the omission errors within burned area products.
Understanding the temporal and spatial patterns of emissions can help stakeholders identify regions for focused mitigation



efforts; therefore, this global cropland burned area product is the first step towards improving global cropland burning emissions.

**Author Contributions**

JH and LG conceptualized the paper. JH developed the algorithm. FA and JH created the reference data sets. MZ helped with quality checking. LG and JR gathered funding. All co-authors contributed to the methodology improvements, provided

resources for deriving GloCAB. JH prepared the manuscript with contributions from all the co-authors.

**Competing Interests**

The authors declare that they have no conflict of interest.

**Acknowledgments**

This work was supported by NASA grants 80NSSC18K0619, 80NSSC18K0739, 80NSSC21K1961, 80NSSC21K1657, and

80NSSC18K0179, and United States Department of the Air Force contract FA8810-18-C-0017. Y. Chen and J. Randerson acknowledge support from NASA's Modeling, Analysis, and Prediction (MAP) program (80NSSC21K1362), SERVIR Applied Sciences Team (80NSSC20K0590), and NASA's Earth Information System (EIS-Fire).

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
