# Peer review of "GloCAB: Global Cropland Burned Area from Mid-2002 to 2020"

_Earth System Science Data, 2023_

## Author Comment (AC1)

**GloCAB: Global Cropland Burned Area from Mid-2002 to 2020**

**Response to RC1**

We thank the Reviewer very much for their comments, corrections, and suggestions. Based on their feedback, we have revised the manuscript and feel that we have improved the context and content of this work. We hope the Reviewer agrees.

**General comments**

The authors present an interesting effort to improve current estimations of burned areas in croplands, which are not well mapped by global BA algorithms, particularly by coarse-resolution sensors. They use a similar approach to GFED4s/5s products, which estimate burned area from active fire information using a set of calibration sites. The study strongly relies on appropriate maps of croplands, which the authors did not produce, neither validate. This is an important gap of the manuscript, since using a coarse resolution global land cover map that includes also errors adds a significant source of uncertainty, as the authors acknowledged, but not quantify. This should be include in the new version of the manuscript.

- We agree with the reviewer that the spatial location of cropland is dependent on the accuracy of the land cover maps. However, this is a known limitation across land cover types and impacts all studies that use land cover maps as an underlying input source. Although the GloCAB product was designed as a stand-alone cropland burned area product, Version 1 was designed to be easily integrated into the upcoming GFED5 product (Chen et al., in review). Our study specifically used the annual MODIS land cover product (classes 12 and 14) to represent cropland area as this allows a seamless merger of GloCAB into the GFED5 product.

The authors base their methodology in MODIS AF, but it is well known that this sensor is close to its functional life. Therefore, I strongly recommend the authors to use VIIRS data instead, or even better from a combination of the two to assure both time series length and future extension. Otherwise, the GloCAB product would just be historical estimation. I realize the convenience of having a long temporal series of crop fires, but this should be balanced with the reduced accuracy from using a coarse resolution AF product.

- GloCAB Version 1 is a historical product by design. It focuses on the ~24-year period of high-quality fire data from MODIS. In a previous study, we created a Ukraine-specific VIIRS-based version of the product (https://iopscience.iop.org/article/10.1088/1748-9326/abfc04). The high-resolution reference maps are the crucial input for the underlying GloCAB methodology and adjustments can be made using alternative crop maps, active fire data, etc. Once the Collection 2 VIIRS active fire product is publicly released, a future combined MODIS/VIIRS version of GloCAB can be created. However, at this point in time, a combined MODIS/VIIRS GloCAB is premature as the VIIRS active fire Collection 1 product has some nontrivial problems

(https://viirsland.gsfc.nasa.gov/PDF/VIIRS_activefire_User_Guide.pdf) that were corrected for Collection 2. Furthermore, GloCAB Version 1 is designed to seamlessly integrate into several emission-based models (e.g. NASA GISS FirePyE model) where previous GFED estimates were used.

- We added a statement to the manuscript highlighting the importance of a future version incorporating MODIS and VIIRS data (lines 73 - 78).

Related to this, the methods do not clearly describe how the AF/BA ratio was applied to just the cropped areas, as within a 0.25° cell many wildfires may occur simultaneously to agricultural burnings. Did you use the cropland area defined by the MOD12 product to mask only the cropped area? What would happen then with the cropped areas where small parcels (< 25 has) are widely extended?

- Yes, to avoid any potential double-counting with burning in adjacent land cover pixels, we filter the MODIS active fire data by pixel-center location within the Class-12 and Class-14 MODIS MCD12Q1 land cover grid cells (Section 2.2). The filtered, presumed-cropland, active fire data were then used to determine which crop type (at the MODIS resolution) was associated with the majority of cropland-filtered active fire pixels for that month. Since the first stages of the analysis occur at the ~500m MODIS grid cell scale, our study assumes that those filtered active fire points are attributed to cropland burning.

In addition, it is not clear either how did the authors extent the conversion factors to other climate regions. Did they apply the winter wheat coefficient to all winter wheat worldwide, regardless the continent-climate zone where they are located?

- We did not take into account different continent-climate zones. For example, we applied the winter wheat conversion factors to all winter wheat pixels globally. Agricultural management practices typically vary by crop type as opposed to climate region, therefore, subdividing conversion factors by climate region will likely not improve the accuracy of the results. A future version of GloCAB could look into this further but would require a large number of reference regions that cover different crop types over different climate zones. For this study, that was not a feasible option.

I appreciate the difficulty of validating the product. However, the exercise that is included in the paper is clearly insufficient to grant any significant confidence to the results. I suggest the authors to generate a few additional agricultural BA maps, similarly to those used for calibration, or at least compare iteratively their results with their calibration sites with a bootstrapping approach.

- Unfortunately, the creation of the high-resolution maps is extremely time-consuming and the publicly available burned area reference datasets (https://essd.copernicus.org/articles/12/3229/2020/) are specifically not designed to map the burned area within cropland. To avoid contamination of our reference data, we purposely avoided reusing our training data as validation data since that would potentially bias our results. If the reviewer would be satisfied with one or potentially two

additional 0.25° mapped grid cells worth of validation, we should be able to accommodate this level of effort (and associated cost) in the time that remains to submit our revised manuscript.

**Specific comments**

Line 48. Perhaps the authors should quote the latest version of GFED.

- The current GFED5 burned area product is under review and the GFED5 emission product is still under development. Hence, we opted to reference the previous GFED product.

Line 68. With the same logic of quoting the algorithm description for the MCD64 product, the authors should quote Lizundia et al., paper for the FireCCI51.

- Thank you for your suggestion. We cited the Lizundia-Loiola paper.

Line 120. Include references of this statement, which is very important as it justifies the selection of the crop types being considered.

- Thank you for your suggestion. We have added references.

Section 2.4 seems more appropriate for the methods section, as it was part of your own developments for the product.

- Thank you for the suggestion. We agree and moved Section 2.4 into the Methods section.

Line 191: "training reference areas, several challenges limited the mapping. Most importantly, small fields and poor air quality in several countries (e.g. India and Thailand) prevented the analysts from observing changes in the fields even with 3-m Planet". This is a bit confusing, since you did not include any calibration site in India or Thailand, according to table 1.

- We had hoped to map reference areas in other parts of the world (e.g. India, Thailand, and Africa), however, the poor air quality and extremely small field sizes in those regions hindered our ability to accurately map these areas. Therefore, the spatial distribution of the reference areas are focused in high-density cropland regions with fields large enough to confidently identify the changing conditions in Planet and Sentinel-2 data. We have added clarity to the sentence (Line 197).

**Response to RC2**

We thank the Reviewer very much for their comments, corrections, and suggestions. Based on their feedback, we have revised the manuscript and feel that we have improved the context and content of this work. We hope the Reviewer agrees.

Emissions from cropland fires. At this stage, the product is not mature enough to generate accurate estimates of GHG emissions from the burning of crop residues. Robust and global Tier I estimates of emissions for the same crops in the paper are already available (e.g. FAOSTAT, EDGAR) based on the area harvested reported from countries. It is recognized that these emissions represent overall a small proportion of agricultural emissions. Yet, the authors decided to open the abstract with a sentence on emissions (from general fires). I would recommend to change the focus in the narrative to other and more important aspects which the product can already address and that the authors already described in the introduction (e.g. use of this information in monitoring systems; health implications; the applications for more sustainable agricultural practices).

- We have made changes in the abstract and introduction to de-emphasize emissions.

Land Cover. The authors used the IGBP land cover type from the MODIS land cover, collection 6. However, the 3 LCCS land cover layers, as reported by Sulla-Menashe and Friedl (2018), are instead the reference type of the MODIS land cover product and those with the higher accuracy. The authors should specify why they decided to use the IGBP type instead. The authors treated the mixed class 14 as full cropland. I tend to agree with this approach given the nature of the data, but it may be worth specifying the reasons for this choice. While I wonder if the authors explored other alternatives (e.g. ESA CCI), I agree that discrete global land cover classifications are all likely to suffer from omission and commission errors and importantly, uneven performances across regions (which the authors should also discuss). Possibly, dynamic land cover products that are better aligned temporally to fire dynamics might be something to look at in future developments. Reference for Sulla-Menashe and Friedl is missing from the bibliography.

- The Collection 6 MCD12Q1 UMD layer had a major error within the land cover classes and the corrected Collection 6.1 product was only available after we started the GloCAB processing. We opted to use the MODIS IGBP Land Cover annual product despite the known inaccuracies as 1) it is widely used with the scientific literature (e.g., Liu et al., 2023; Van Der Werf, 2017), 2) it would allow GloCAB to easily be incorporated into GFED5, and 3) the MODIS land cover product (as opposed to the ESA CCI land cover product) is most compatible with the MODIS Active Fire data (e.g. sinusoidal projection, spatial resolution, annual product).
- We have added the updated Sulla-Menashe and Friedl reference.

  References

- Liu, P., Liu, Y., Guo, X., Zhao, W., Wu, H., & Xu, W. (2023). Burned area detection and mapping using time series Sentinel-2 multispectral images. *Remote Sensing of Environment*, *296*, 113753.

- Van Der Werf, G. R., Randerson, J. T., Giglio, L., Van Leeuwen, T. T., Chen, Y., Rogers, B. M., ... & Kasibhatla, P. S. (2017). Global fire emissions estimates during 1997–2016. *Earth System Science Data*, *9*(2), 697-720.

Cropland fires. The authors have included some information on the characteristics of cropland fires. However, I believe it would be beneficial to present this in a dedicated section discussing the main differences between pre-planting, pre-harvesting and post-harvesting fires; associations with crop type (e.g. pre-harvesting fires in sugar cane fields) and agricultural practices as well as the implications for detection (e.g. pre-planting fires are followed rather rapidly by soil preparation which alters the burned area and reduce the ability to detect the area that was burnt). Information on the prevalence of geographical distribution by crop type and gaps in literature would be also useful.

- We have added some additional information to the introduction.

Trends and Africa. The abstract does not discuss the role of Africa in the assessment of global trends and the manuscript presents global results with and without Africa only in the supplementary section. This region certainly contributes significantly to the uncertainty of this product, including due to the prevalent small size of crop fields in the region. A more structured discussion of the caveats of their approach in Africa and the impact of this region for the global

- We have added some additional information to the manuscript. Within all land cover types, Africa accounts for 67% of the burned area, therefore it has a considerable impact on "global" burned area statistics. However, within croplands (as defined by MCD12Q1), Africa only accounts for 27% of the burned area. We included the supplementary figure to show that the trends do not change significantly when removing Africa. We have added some additional text to the manuscript too.

**Response to RC 3**

We thank the Reviewer very much for their comments, corrections, and suggestions. Based on their feedback, we have revised the manuscript and feel that we have improved the context and content of this work. We hope the Reviewer agrees.

**Major comments**

The GloCAB dataset offers the first global cropland-focused burned area product, providing monthly data from 2002 to 2020 at 0.25° resolution, targeting small-fire types in croplands. This first-of-its-kind effort offers a detailed view of small-fire types in croplands, highlighting region-specific trends and paving the way for improved understanding and policy-making. I would like to suggest a few more things that need to improve in this study.

The author can highlight the novelty of cropland-focused burned area mapping further. In the current version, it is difficult to see a significant difference from previous products. It would be nice to have more discussion about the implications that only GloCAB has.

- We have added additional clarity on the novelty and implications of GloCAB in Section 1.

Also, as the land cover product influences the overall results, the author quantifies the uncertainty of cropland-burned area mapping using different land cover datasets.

- We agree that the choice of land cover product will influence the overall burned area result, especially within cropland regions (e.g., Zubkova et al., 2023). Ideally, future research should focus on improving these crucial underlying input layers (e.g., land cover products). Within the scope of our project and in particular, with our limited funding, we chose to focus on developing the cropland burned area reference data and burned area algorithm. The MODIS IGBP Land Cover annual product is widely used (e.g., Liu et al., 2023; van der Werf, 2017), and although there are known inaccuracies, we chose to use the land cover product that would allow GloCAB to easily be incorporated into GFED5.

References

- Zubkova, M., Humber, M. L., & Giglio, L. (2023). Is global burned area declining due to cropland expansion? How much do we know based on remotely sensed data?. *International Journal of Remote Sensing*, *44*(4), 1132-1150.
- Liu, P., Liu, Y., Guo, X., Zhao, W., Wu, H., & Xu, W. (2023). Burned area detection and mapping using time series Sentinel-2 multispectral images. *Remote Sensing of Environment*, *296*, 113753.
- van der Werf, G. R., Randerson, J. T., Giglio, L., Van Leeuwen, T. T., Chen, Y., Rogers, B. M., ... & Kasibhatla, P. S. (2017). Global fire emissions estimates during 1997–2016. *Earth System Science Data*, *9*(2), 697-720.

**Minor comments**

Line 18-20: I agree with the author's point about cropland-focused burned area mapping. But this sentence does not link smoothly with the previous context

- We have edited the sentence to improve the link with the previous context.

Line 73-75: As MODIS will be decommissioned, VIIRS will be used alternatively. For long-term analysis, intercalibrating between MODIS and VIIRS is needed. That information could be useful for readers.

- We have added a short statement on the need to intercalibrate between MODIS and VIIRS (lines 73 - 78)

Method: I think the reliability of GloCAB heavily relies on the performance of MODIS products. The author can simply add quantified uncertainty of each input dataset.

- Adding a useful uncertainty estimate is not feasible as it will require more high-resolution reference data. Furthermore, the input datasets have very different measures of uncertainty: the active fire product uses false alarm rates, while the land cover product uses misclassification rates. Neither of these are useful measures for end users when using GloCAB (burned area).

Figure 7, 8, and 9 should be improved with clear color and formal legend. Statistical values also can be included in the figure.

- Thank you for your suggestion. We made adjustments to each figure, including lightening up the grey so the colors stand out.

---

## Author Response (AR2)

**GloCAB: Global Cropland Burned Area from Mid-2002 to 2020**

**Response to Topical Editor**

We thank the Topical Editor for reviewing our paper. We have adjusted the sentence to improve the clarity. All references were edited to meet the journal requirements and one of the figures (Figure 1) was adjusted based on the Color Blindness Simulator. The other figures were suitable based on the Simulator.